# Effect of Cow Manure Biochar on Lettuce Growth and Nitrogen Agronomy Efficiency

**DOI:** 10.3390/plants13233326

**Published:** 2024-11-27

**Authors:** Jae-Hyuk Park, Han-Na Cho, Ik-Hyeong Lee, Se-Won Kang

**Affiliations:** 1Department of Agricultural Chemistry, Sunchon National University, Suncheon 57922, Republic of Korea; jaehyuk0309@naver.com (J.-H.P.); hbabyn@hanmail.net (H.-N.C.); jason1384@naver.com (I.-H.L.); 2Department of Agricultural Life Sciences, Sunchon National University, Suncheon 57922, Republic of Korea

**Keywords:** lettuce growth, nitrogen use efficiency, upland soil, cow manure biochar, crop productivity

## Abstract

This study aimed to produce livestock manure biochar to decrease environmental problems from livestock manure and evaluate its effectiveness as an organic fertilizer by examining the growth and nutrient use efficiency of crops. A plot experiment was conducted to investigate the characteristics of lettuce growth and nitrogen use efficiency in upland soils treated with cow manure biochar. The cow manure biochar was applied at rates of 0, 3, 5, 7, and 10 t ha^−1^ (referred to as CMB0, CMB3, CMB5, CMB7, and CMB10, respectively), along with inorganic fertilizer (IF, NPK—200-59-12 kg ha^−1^). The lettuce cultivation test was carried out for 42 days, during which the fresh weight, dry weight, length, and number of lettuce leaves were measured. Nitrogen use efficiency was evaluated by determining the agronomic efficiency of N and the apparent recovery fraction of N. Overall, as the cow manure biochar application rate increased, crop growth and nitrogen uptake improved. Soils treated with CMB5 and CMB7 showed higher lettuce growth, nitrogen content, and nitrogen uptake compared to soils under other treatments. Nitrogen use efficiency followed a pattern similar to that of crop productivity, with cow manure biochar application levels playing a significant role. In particular, the agronomic efficiency of N and the apparent recovery fraction of N, which are both related to crop nutrient utilization, were significantly higher in the CMB5 treatment compared to the IF treatment. These results indicate that nitrogen use efficiency can be enhanced through biochar application when growing crops on agricultural land. Therefore, it is suggested that the appropriate application of cow manure biochar can reduce inorganic fertilizer use and increase crop productivity, thereby enabling sustainable and eco-friendly agriculture.

## 1. Introduction

Manure is a byproduct of livestock production and contains organic matter and nutrients that can enhance soil health by increasing nutrient cycling, nutrient retention, and water-holding capacity [1,2]. It can reduce production costs by partially or completely replacing inorganic fertilizers by providing essential nutrients to crops. When used appropriately as a crop fertilizer or processed into a byproduct, livestock manure is a valuable resource [3,4,5].

Traditionally, livestock manure has been applied as a soil fertilizer and is still widely used in this manner. In South Korea, there are two main methods for managing livestock manure: purification and discharge, and resource recovery. With regards to resource recovery, over 90% of the manure is recycled as compost or liquid fertilizer and returned to farmland. However, as the scale of the livestock industry has grown and the number of livestock has increased, the amount of livestock waste generated has also risen, creating a serious waste management challenge [6,7]. Livestock manure is known to cause significant environmental issues, such as green algae and eutrophication, when it flows into water systems [8,9]. Also, manure application can cause antibiotic-resistant genes due to antibiotic residues in the environment and potential contamination by heavy metals [10]. Thus, research on effective livestock waste treatment methods has become increasingly necessary.

In South Korea, the demand for organic agricultural products has steadily increased as consumers have become more interested in safe food. In the meantime, compost and liquid fertilizer are being produced through the recycling of livestock manure, and the produced compost and liquid fertilizer are returned to farmland to perform crop production and soil management through eco-friendly management [11,12]. However, the amount of livestock manure compost and liquid fertilizer applied to farmland has been gradually decreasing due to the excessive accumulation of nutrients in the soil. Additionally, due to the carbon neutrality policy, further utilization of livestock manure is required to address the greenhouse gases emitted during the production and application of livestock manure compost and liquid fertilizer [13].

Biochar has been shown to be effective in addressing climate change, energy production, soil improvement, and waste management. In addition to enhancing soil fertility, biochar increases crop yields, reduces greenhouse gas emissions, and decreases pollutant levels, offering numerous benefits when applied to soil [14,15]. Additionally, biochar has emerged as a promising material for carbon neutrality and agricultural land management [16]. It is a carbon-rich substance produced by the pyrolysis of organic biomass under oxygen-free conditions [17]. Research on biochar has primarily focused on its potential to reduce greenhouse gas emissions, improve crop growth, enhance soil management, and adsorb pollutants. Most studies have used biochar produced from woody and herbaceous biomass sources [18,19,20].

Meanwhile, few studies have examined the use of livestock manure biochar globally because converting livestock manure into biochar requires significant time and cost. However, converting livestock manure into biochar is considered an ideal method for environmentally friendly farmland management due to its potential to address the environmental problems caused by livestock manure [21]. This study aimed to evaluate the effects of livestock manure biochar as an organic fertilizer on crops and to provide fundamental data for its future application in sustainable agriculture. Therefore, the objective of this study was to evaluate the effects of livestock manure biochar application on lettuce growth, soil pH, SOC, and nitrogen use efficiency. The results obtained in this study can be applied to the methods of recycling livestock manure for controlling environmental issues and the management of upland cultivation in the future.

## 2. Materials and Methods

### 2.1. Raw Soil

The soil used for crop cultivation was collected from an upland area in Jeongeup City, South Korea (Table 1). The soil had 5.24 for pH and 0.63 dS m^−1^ for EC. The organic matter (OM), total nitrogen (TN), and available phosphate (Avail. P_2_O_5_) of raw soil were 15.3 g kg^−1^, 1.45 g kg^−1^, and 209 mg kg^−1^, respectively. The soil contains 0.22 cmol_c_ kg^−1^ for K, 5.61 cmol_c_ kg^−1^ for Ca, 1.34 cmol_c_ kg^−1^ for Mg, and 7.82 cmol_c_ kg^−1^ for CEC.

### 2.2. Preparation of Cow Manure Biochar

In this study, completely dried cow manure was used as the raw material to gain the biochar. The cow manure biochar (CMB) was produced at Sunchon National University in a stainless-steel container with a cover under oxygen-limited conditions and was pyrolyzed in a furnace (DK-1015(E), STI tech, Gumi, Republic of Korea). The condition of CMB production was through pyrolysis of 400 °C for 2 h using equipment with anaerobic condition and injection flow of nitrogen gas (Figure 1).

The characteristics of the CMB are listed in Table 1. The yield of the CMB was 42.3%, and it contained pH of 9.36, nitrogen (N) of 2.12%, phosphorus (P_2_O_5_) of 0.79%, and potassium (K_2_O) of 2.98%. The C, H, and O contents of CMB were 37.3, 1.95, and 13.7%, respectively (Table 2).

### 2.3. Experimental Design

The selected crop was lettuce (*Lactuca sativa* L.). The experiment was conducted in 2024 by applying CMB to the experimental plots (1 mL × 1 mW × 0.5 mH) in a glasshouse with controlled temperature and humidity conditions. Lettuce cultivation was carried out in a randomized block design with three replications, and the seedlings were transplanted into each plot with 25 cm spacing. The air temperature and relative humidity in the glasshouse were maintained at 22/18 °C (day/night) and 65%, respectively. Growth characteristics, including fresh weight, dry weight, leaf length, and leaf number, were measured after 42 days (16 February–29 March).

CMB was applied to the plot cultivation at 0, 3, 5, 7, and 10 t ha^−1^ (CMB0, CMB3, CMB5, CMB7, and CMB10, respectively), and an inorganic fertilizer (IF) treatment was used as a control. The amounts of N-P_2_O_5_-K_2_O fertilizers applied were 200-25-128 kg ha^−1^ according to soil management and fertilizer recommendations provided by the Rural Development Administration (RDA) of South Korea.

### 2.4. Nitrogen Analysis

The collected lettuces were dried in an oven at 70 °C for 3 days, after which they were digested with H2_S_O_4_+H_2_O_2_. The TN analysis of the prepared plants was performed by applying the Kjeldahl method.

The nitrogen use efficiency of lettuce under different CMB applications was calculated by the following equation and approached from two aspects [22].
The agronomic efficiency of N (AE_N) = (Y − Y_0_)/Input N(1)
The apparent recovery fraction of N (ARF_N) = (Upt − Upt_0_)/Input N(2)

In the equations, Y represents the yield obtained in the treatment with nitrogen (N) application, Y_0_ is the yield observed in the treatment without any N application, Upt denotes the N taken up by the crop in the treatment with N application, Upt_0_ indicates the N taken up by the crop in the treatment without N application, and input N refers to the total amount of nitrogen supplied through the CMB and inorganic N fertilizer applied (kg ha^−1^).

### 2.5. Sample Analysis

The pH and EC were measured at a soil–water ratio of 1:5 after shaking the mixture for 30 min. The OM and TN analyses were performed using the Tyurin and Kjeldahl methods, respectively. Available P_2_O_5_ was measured using the Lancaster method and exchangeable cations in the soil were extracted by 1N-NH_4_OAc. Soil analysis was performed as described by NIAST [23].

The CMB was analyzed for pH, N, P_2_O_5_, K_2_O, and elemental composition (C, H, and O). The pH was measured at a soil–water ratio of 1:10 after shaking the mixture for 30 min. N, P_2_O_5_, and K_2_O were measured using the wet digestion method (H_2_SO_4_+HClO_4_), and the elemental composition of CMB was analyzed using an organic elemental analyzer (FlashSmart, ThermoFisher Scientific, Seoul, Republic of Korea).

### 2.6. Statistical Analysis

Statistical analyses were performed using a statistical analysis system (SPSS Statistics Version 27). The mean values were calculated as the averages of three replicates. Each mean value was subjected to an analysis of variance (ANOVA), and comparisons of the treatments were performed using Duncan’s multiple range test (DMRT) at a 5% probability level.

## 3. Results

### 3.1. Growth Characteristics of Lettuce by Cow Manure Biochar Applications

The investigation results related to the growth characteristics of lettuce after CMB application are shown in Table 3. Overall, changes in lettuce growth were influenced by the application of CMB. Among the growth characteristics of lettuce, the fresh weight (leaf + root) of lettuce was 56.5, 65.9, 81.3, 81.6, 75.3, 75.3, and 79.9 g plant^−1^ in CMB0, CMB3, CMB5, CMB7, CMB10, and IF treatments, respectively, indicating differences between the CMB application levels. Leaf length was not significantly different among the CMB treatments, but it was confirmed that the mean value was higher in the CMB5 treatment. The number of lettuce leaves in the CMB treatment was significantly higher than in the CMB0 treatment.

### 3.2. Nitrogen Contents of Lettuce Plants by Cow Manure Biochar Applications

Table 4 shows the changes in dry weights, TN content, and N uptake following N application rates and after crop harvesting under different plant parts of the lettuce. Overall, the dry weight had a greater effect on nitrogen uptake compared to nitrogen content and showed significant results in lettuce cultivation. Additionally, the characteristics of each part of the lettuce showed higher nitrogen uptake in the leaves of lettuce, with a higher dry weight compared to the roots. Irrespective of the treatment conditions, nitrogen uptake in different parts of the lettuce was higher in the order of leaves > roots. The dry weights of leaves in lettuce were higher in the order of IF ≒ CMB5 > CMB7 > CMB10 > CMB3 > CMB0 treatments. The dry weights of the lettuce roots were similar to those of the leaves. The TN contents of leaves and roots in lettuce that had undergone CMB applications were shown to be 3.45~4.03% and 2.55~3.62%, respectively, and when CMB application was 5t ha^−1^ or more, the TN contents were similar to those of the IF treatment. The nitrogen uptake of leaves and roots in lettuce in the CMB5 treatment was 4.70 g m^−2^ and 0.40 g m^−2^, respectively, which were all higher than that in other CMB treatments, indicating a significant difference according to CMB application levels.

### 3.3. Nitrogen Use Efficiency of Lettuce Plants by Cow Manure Biochar Applications

The change in nitrogen use efficiency during lettuce cultivation varied significantly with the CMB application (Figure 2). Lettuce cultivation resulted in a higher agronomic efficiency of N (AE_N) and apparent recovery fraction of N (ARF_N) in the CMB treatments, excluding CMB10, compared to the IF treatment. The AE_N of lettuce in the CMB5 treatment was more than 2 kg N ha^−1^ higher than that of the IF treatment, and the ARF_N in the CMB3, 5, and 7 treatments was higher than that of the IF treatment. These results show that CMB application helps absorb nitrogen in crops.

### 3.4. Changes of pH and SOC by Cow Manure Biochar Applications

Figure 3 shows changes of pH and SOC by CMB applications after lettuce harvesting. Soil pH ranged between 5.14 and 5.49 and SOC content ranged between 8.15 and 11.36 g kg^−1^. The pH and SOC content after CMB applications increased by 0.10–0.18 and 0.90–2.40 g kg^−1^. Totally, soil pH and SOC content in CMB treatments was noticeably higher after lettuce harvesting compared to raw soil.

## 4. Discussion

### 4.1. Growth and Nitrogen Uptake of Lettuce by Cow Manure Biochar

The results of this study show that CMB application had a positive effect on lettuce growth. Unlike biochars made from general woody and herbaceous biomass, biochar from livestock manure has a high nutrient content, so it was possible to achieve effects equivalent to those of fertilizer treatment. In particular, in terms of lettuce growth, it was demonstrated that when CMB was applied at 5–7 t ha^−1^, the growth of lettuce was similar to or higher than that of the IF treatment. There was no significant difference in the nitrogen content of lettuce across different parts between the CMB5-10 and IF treatments. The amounts of nitrogen uptake in lettuce by parts were greatly influenced by dry weight, with the CMB5 and IF treatments showing higher values compared to the other treatments. The effects of biochar on crop growth improvement have been reported in previous studies. According to Singh et al. [24], crop growth is enhanced when biochar’s high pH and CEC enhance soil nutrient availability. Similarly, Burrell et al. [25] reported that biochar application improves soil density and porosity, while Sohi et al. [26] noted that its high carbon content provides excellent stability in soil. In addition to these effects, biochar application is known to be more effective in improving crop growth in mixed treatments of biochar and other nutrients than in single biochar applications [27,28,29]. In biochar research, inorganic fertilizers or organic substances are often combined to improve growth and increase crop yields due to the low nutrient content of biochar [30,31]. Additionally, Xu et al. [32] reported that a mixed treatment of biochar and fertilizer affected SOC and the C:N ratio, influencing crop growth. These results suggest that combining biochar with fertilizer components can effectively increase crop yields. Therefore, considering waste recycling, stable crop production, and carbon neutrality, crop cultivation using CMB with various nutrients may be more effective than using woody and herbaceous biochars and is expected to have high potential for use in the agricultural sector.

### 4.2. Nitrogen Use Efficiency by Cow Manure Biochar

As investigated in this study, the growth and nitrogen uptake of lettuce changed significantly depending on the level of CMB application. Nitrogen is one of the essential inputs in agriculture because it is an element with high plant nutrient requirements and can either increase or limit crop productivity [33,34,35]. Therefore, improving nitrogen use efficiency can prevent excessive fertilizer application and accumulation, thereby helping to reduce environmental pollution [28,36]. Stoykova et al. [37] and Xia et al. [34] reported that the appropriate application of biochar and fertilizer components during crop cultivation can enhance nitrogen use efficiency. In a paddy-upland rotation study, Zhang et al. [38] reported that the soil incorporation of biochar not only improved the nitrogen use efficiency of crops but also reduced greenhouse gas emissions. These findings align with the results of this study, showing increased nitrogen use efficiency in lettuce when livestock manure biochar containing nutrients was applied. In the treatment area where fertilizer components were combined with biochar, it was reported that biochar improved soil physical properties to facilitate nutrient flow and enhanced chemical properties, such as SOC, to promote nitrogen utilization [39,40]. It has also been reported that the activities of microorganisms and enzymes related to nitrogen were involved in nitrogen use efficiency [34,37]. Meanwhile, Gu et al. [41] reported that nitrogen use efficiency decreases as nitrogen input increases. Similarly, Han et al. [42], who investigated nitrogen use efficiency through a meta-analysis related to the input of biochar and fertilizer, also showed a decrease in nitrogen use efficiency. Consistent with this study, AE_N and ARF_N were low in the CMB10 treatment group, where nitrogen input was the highest. In conclusion, the application of livestock manure biochar in crop cultivation can further improve the nitrogen absorption and use efficiency of crops, promote crop growth, increase crop yield, and reduce environmental pollution.

### 4.3. Soil pH and SOC by Cow Manure Biochar

Soil pH and SOC are important factors that affect crop growth, soil properties, and microbial communities, and they can broadly affect greenhouse gas emissions [32,43]. In this study, the pH and SOC of the CMB treatment groups were significantly different from those of the CMB0 and IF treatments. The pH and SOC of each treatment condition tended to increase as the application level of CMB increased. The high pH and carbon content of biochar could affect soil pH and SOC, which was consistent with previous studies [44,45]. Thus, conversion of livestock manure into biochar was considered to have a positive effect on crop growth, nitrogen use efficiency, and soil properties.

## 5. Conclusions

Overall, the appropriate application of CMB has been shown to have positive effects on lettuce growth and nitrogen use efficiency in upland environments. In particular, the 3–7 t ha⁻^1^ CMB treatments exhibited nitrogen use efficiencies equal to or higher than those in the AE_N and ARF_N treatments, despite receiving lower nitrogen inputs compared to the IF treatment. These results indicate that CMB can enhance crop growth and nitrogen use efficiency without the need for additional nitrogen fertilizer input. Moreover, no symptoms of damage caused by CMB were found during the lettuce growing period, and CMB application can replace inorganic fertilizers, resulting in cost savings. Additionally, the CMB application was investigated to be effective in improving the pH and SOC of lettuce cultivation soil. Therefore, considering the other benefits of biochar application, such as recycling livestock manure, improving crop productivity and soil fertility, enhancing nutrient use efficiency, and replacement of inorganic fertilizers, we propose an appropriate level of CMB application in fields.

## Figures and Tables

**Figure 1 plants-13-03326-f001:**
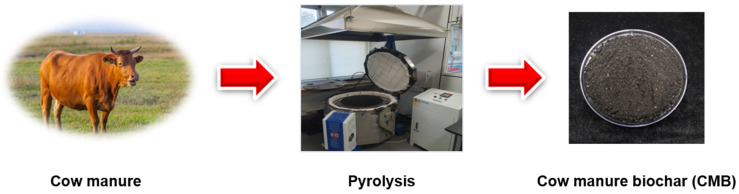
Production of cow manure biochar using furnace equipment.

**Figure 2 plants-13-03326-f002:**
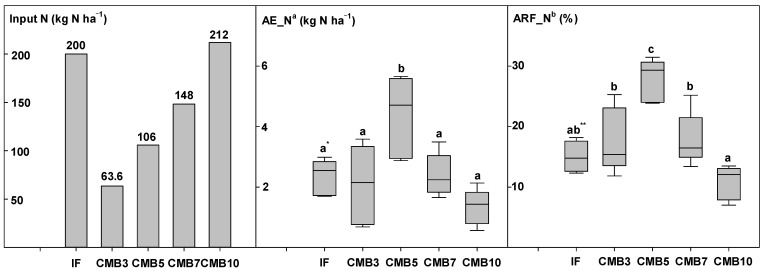
Nitrogen use efficiency in lettuce production as influenced by cow manure biochar application: AE_N^a^, agronomic efficiency of N (kg yield increase per kg input N); ARF_N^b^, apparent recovery fraction of N (kg N uptake increase per kg input N); *, ** Different letters indicate significant differences among treatments at the 5% probability level.

**Figure 3 plants-13-03326-f003:**
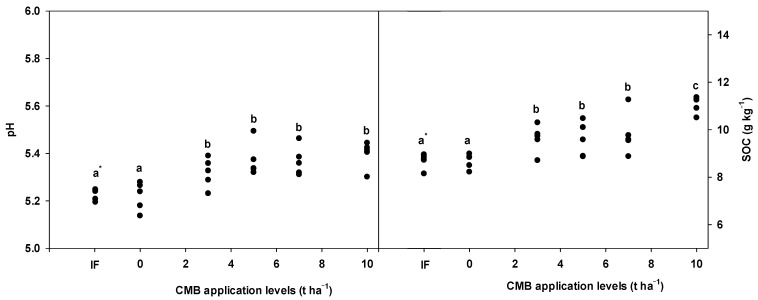
Changes of pH and SOC after lettuce harvesting: * Different letters indicate significant differences among treatments at the 5% probability level.

**Table 1 plants-13-03326-t001:** Properties of experimental soil used in this study.

pH	EC	OM	TN	Avail. P_2_O_5_	Ex. Cations (cmol_c_ kg^−1^)			
(1:5H_2_O)	(dS m^−1^)	(g kg^−1^)		(mg kg^−1^)	K	Ca	Mg	CEC
5.24 ± 0.10	0.63 ± 0.06	15.3 ± 0.78	1.45 ± 0.11	209 ± 11.4	0.22 ± 0.01	5.61 ± 0.18	1.36 ± 0.02	7.82 ± 0.04

**Table 2 plants-13-03326-t002:** Characteristics of biochar derived from cow manure.

Yield	pH	N	P_2_O_5_	K_2_O	Elemental Composition			Molar Ratio	
					C	H	O	H/C	O/C
(%)	(1:10H_2_O)	- - - - - - - - (%) - - - - - - - -			- - - - - - - - (%) - - - - - - - -				
42.3 ± 0.35	9.36 ± 0.09	2.12 ± 0.03	0.79 ± 0.08	2.98 ± 0.07	37.3 ± 0.58	1.95 ± 0.01	13.7 ± 0.71	0.63 ± 0.01	0.28 ± 0.02

**Table 3 plants-13-03326-t003:** Growth characteristics of lettuce by cow manure biochar application.

Treatment	Fresh Weight (g plant^−1^)		Leaf Length	Leaf Number
	Leaf	Root	(cm plant^−1^)	(ea plant^−1^)
CMB0	53.8 a *	2.70 a	11.3 a	23.6 a
CMB3	61.6 b	4.34 c	13.5 b	27.8 ab
CMB5	76.5 d	4.77 d	14.8 b	28.0 ab
CMB7	76.7 d	4.85 d	13.9 b	28.6 ab
CMB10	71.9 c	3.42 b	14.4 b	27.4 b
IF	75.4 cd	4.51 cd	14.8 b	29.6 c

* Different letters within a column are not significantly different at a probability level of 0.05, according to Duncan’s multiple-range test.

**Table 4 plants-13-03326-t004:** Characteristics of dry weight and nitrogen contents in lettuce.

Part	Treatment	Dry Weight	TN Content	Nitrogen Uptake
		(g m^−2^)	(%)	(g m^−2^)
Leaf	CMB0	78.7 a *	2.61 a	2.05 a
	CMB3	87.4 ab	3.45 b	3.02 b
	CMB5	119.4 c	3.94 c	4.70 cd
	CMB7	109.2 bc	4.03 c	4.41 cd
	CMB10	105.7 b	3.96 c	4.20 c
	IF	120.7 c	3.98 c	4.81 d
Root	CMB0	5.9 a	1.72 a	0.10 a
	CMB3	10.4 c	2.55 b	0.26 b
	CMB5	11.3 d	3.49 cd	0.40 d
	CMB7	10.9 cd	3.62 d	0.39 d
	CMB10	7.2 b	3.39 c	0.24 b
	IF	10.5 cd	3.34 c	0.35 c

* Different letters within a column are not significantly different at a probability level of 0.05, according to Duncan’s multiple-range test.

## Data Availability

All data are contained within the article.

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
