# Peer review of "Effect of Cow Manure Biochar on Lettuce Growth and Nitrogen Agronomy Efficiency"

_plants, 2024, doi:10.3390/plants13233326_

Round 1

Reviewer 1 Report

Comments and Suggestions for Authors

The manuscript titled ‘’ Effect of Cow Manure Biochar on Lettuce Growth and Nitrogen 2

Agronomy Efficiency’ faced with some problem which needs to be modified before further process. Few of such drawbacks were mentioned below:

1. The novelty of this work is uncertain; please make it more clear and broad

2. Don’t use the abbreviation in abstract  

3. Agriculture word is not suitable in keywords after lettuce growth, so replace this word  

4. Line 29 to 30… needed 1 reference at least; this paper actually can make confidence of your statement (https://doi.org/10.1016/j.chemosphere.2024.142368)

5. Add more references between Line 55 to 59, I will suggest some papers must read and can enhance the sentence value   https://doi.org/10.1186/s12870-024-05058-9; https://doi.org/10.1080/01904167.2021.1871746

6. Must draw table for soil properties in Materials and Methods section

7.   Line 105, should write correct unit kg ha-1 instead of kg ha-1

8. Add more recent references in discussion section

9. Improve the conclusion section according to your findings

10. Should improve the figures resolution and clarity  

Comments on the Quality of English Language

The English could be improved to more clearly express the research.

Author Response

Response to Reviewer 1 Comments

The manuscript titled ‘’ Effect of Cow Manure Biochar on Lettuce Growth and Nitrogen Agronomy Efficiency’ faced with some problem which needs to be modified before further process. Few of such drawbacks were mentioned below:

Response : Thank you for your valuable comments and suggestions. We greatly appreciate the time and effort from the subject editor and reviewer for reviewing our manuscript.

  1. The novelty of this work is uncertain; please make it more clear and broad

Response : Thanks for your valuable comments and suggestions. We have now added novelty from suggested your opinion in the “manuscript” and cleary them accordingly. Further, we made other minor changes in the manuscript for novelty and clarity.

  1. Don’t use the abbreviation in abstract

Response : Agreed and changed

  1. Agriculture word is not suitable in keywords after lettuce growth, so replace this word

Response : Agreed and changed to “crop productivity”.

  1. Line 29 to 30… needed 1 reference at least; this paper actually can make confidence of your statement (https://doi.org/10.1016/j.chemosphere.2024.142368)

Response : Thank you for pointing out this. We have added related references as suggested.

  1. Add more references between Line 55 to 59, I will suggest some papers must read and can enhance the sentence value https://doi.org/10.1186/s12870-024-05058-9; https://doi.org/10.1080/01904167.2021.1871746

Response : Thank you for pointing out this. We have added related references as suggested. "https://doi.org/10.1186/s12870-024-05058-9; https://doi.org/10.1080/01904167.2021.1871746"

  1. Must draw table for soil properties in Materials and Methods section

Response : According to reviewer’s comment, I added the Table.

  1. Line 105, should write correct unit kg ha-1 instead of kg ha-1

Response : Agreed and changed to “kg ha-1

  1. Add more recent references in discussion section

Response : Thank you; some of the following papers has now been cited in the text wherever appropriate.

Jiang, Y.; Li, T.; Xu, X.; Sun, J.; Pan, G.; Cheng, K. A global assessment of the long-term effects of biochar application on crop yield. Curr. Res. Environ. Sustain. 2024, 7, 100247. https://doi.org/10.1016/j.crsust.2024.100247.

Sadra, S.; Mohammadi, G.; Mondani, F. Effects of cover crops and nitrogen fertilizer on greenhouse gas emissions and net global warming potential in a potato cropping system. J. Agric. Food Res. 2024, 18, 101256. https://doi.org/10.1016/j.jafr.2024.101256.

  1. Improve the conclusion section according to your findings

Response : Modified as suggested. Some part has been rewritten in conclusion section.

  1. Should improve the figures resolution and clarity

Response : Thanks for your comment. I currently have a manuscript file that I can't find any broken parts in the picture. If you tell me specifically, I will fix it.

Thanks for your positive review, again.

I am looking forward to seeing your positive evaluation soon.

Best regard

Se-Won Kang

Reviewer 2 Report

Comments and Suggestions for Authors

The manuscript titled ‘’Impact of Cow Manure Biochar Application on Lettuce Growth and Nitrogen Use Efficiency: Recycling Cow Manure to Mitigate Environmental Issues’’ is good research dealing with manure-based biochar and its effectiveness on plant growth. Overall, the logic of the study makes sense but I’m a bit questioning about the novelty of the work! There are many publications that have used manure biochar for agricultural proposes like organic fertilization. Now, changing the type of plant in the experiments doesn't guarantee new insights into the topic. In general, the manuscript is well structured and has sufficient discussion. If the author could boost the novelty of the work (clearly) it may be considered for further processing. There are some points below as well:

The abstract does not have a good beginning! The authors directly started with the experimental design! Please clearly state the main concerns that made you decide to do this research. What is the knowledge gap? (in 2-3 lines).

Lines 49-53: ‘’However, the amount of live-49 stock manure compost and liquid fertilizer applied to farmland has been gradually decreasing due to the excessive accumulation of nutrients in the soil. Additionally, due to the carbon neutrality policy, further utilization of livestock manure is required to address the greenhouse gases emitted during the production and application of livestock manure compost and liquid fertilizer.‘’ -> please provide references for these lines!

Lines 55-58: ‘’Biochar has been shown to be effective in addressing climate change, energy production, soil improvement, and waste management. In addition to enhancing soil fertility, biochar increases crop yields, reduces greenhouse gas emissions, and decreases pollutant levels, offering numerous benefits when applied to soil. ‘’ -> Please avoid general talking and use related literature to boost the reliability of your statements. Please bring at least one ref in () in front of each benefit. I would suggest the following ref to address mitigating climate change, but please find some more by yourself: https://doi.org/10.1002/ldr.4006

Last paragraph of the introduction: The ‘’novelty’’, ‘’objectives’’, and ‘’hypothesis’’ of the work are not clear! Please clearly state these important elements there!

Line 72: Raw soil?

Lines 87-90: Please briefly describe what method and devices were used for biochar analysis! (company and country of origin for devices as well!)

In conclusion, please avoid repeating the results again here. Just mention the most important ones and give your final suggestions based on the findings.

Author Response

Response to Reviewer 2 Comments

The manuscript titled ‘’Impact of Cow Manure Biochar Application on Lettuce Growth and Nitrogen Use Efficiency: Recycling Cow Manure to Mitigate Environmental Issues’’ is good research dealing with manure-based biochar and its effectiveness on plant growth. Overall, the logic of the study makes sense but I’m a bit questioning about the novelty of the work! There are many publications that have used manure biochar for agricultural proposes like organic fertilization. Now, changing the type of plant in the experiments doesn't guarantee new insights into the topic. In general, the manuscript is well structured and has sufficient discussion. If the author could boost the novelty of the work (clearly) it may be considered for further processing. There are some points below as well:

Response : Thanks for your positive comments. Reviewer's comments were immensely helpful to improve the quality of the paper. We have revised the manuscript in response to reviewer’s comment. We have incorporated all the suggestions of the reviewer.

  1. The abstract does not have a good beginning! The authors directly started with the experimental design! Please clearly state the main concerns that made you decide to do this research. What is the knowledge gap? (in 2-3 lines).

Response : Agreed and added to “This study aimed to produce livestock manure biochar to decrease environmental problems from livestock manure, and evaluate its effectiveness as an organic fertilizer by examining the growth and nutrient-use efficiency of crops”.

  1. Lines 49-53: ‘’However, the amount of live-49 stock manure compost and liquid fertilizer applied to farmland has been gradually decreasing due to the excessive accumulation of nutrients in the soil. Additionally, due to the carbon neutrality policy, further utilization of livestock manure is required to address the greenhouse gases emitted during the production and application of livestock manure compost and liquid fertilizer.‘’ -> please provide references for these lines!

Response : Thank you for pointing out this. We have added related reference as suggested. " https://doi.org/10.7745/KJSSF.2024.57.4.253"

  1. Lines 55-58: ‘’Biochar has been shown to be effective in addressing climate change, energy production, soil improvement, and waste management. In addition to enhancing soil fertility, biochar increases crop yields, reduces greenhouse gas emissions, and decreases pollutant levels, offering numerous benefits when applied to soil. ‘’ -> Please avoid general talking and use related literature to boost the reliability of your statements. Please bring at least one ref in () in front of each benefit. I would suggest the following ref to address mitigating climate change, but please find some more by yourself: https://doi.org/10.1002/ldr.4006

Response : Thank you for pointing out this. We have added related references as suggested. "https://doi.org/10.1186/s12870-024-05058-9; https://doi.org/10.1080/01904167.2021.1871746"

  1. Last paragraph of the introduction: The ‘’novelty’’, ‘’objectives’’, and ‘’hypothesis’’ of the work are not clear! Please clearly state these important elements there!

Response : introduction section has been revised to offer a clear meaning. The objective is made clear. And adequate background is provided.

  1. Line 72: Raw soil?

Response : This is the soil used in the lettuce cultivation experiment. Raw soil refers to the soil before the experiment.

  1. Lines 87-90: Please briefly describe what method and devices were used for biochar analysis! (company and country of origin for devices as well!)

Response : Thanks for your opinion. Information related to the analysis device was added in the "2.2. and 2.5. section".

2.2 “The Cow manure biochar (CMB) was produced at Sunchon National University in a stainless container with a cover under oxygen-limited conditions and pyrolyzed in a fur-nace (DK-1015(E), STI tech, Gumi, Republic of Korea). The condition of CMB production was through pyrolysis of 400°C for 2 h using equipment with anaerobic condition and in-jection flow of nitrogen gas (Figure 1).”

2.5. The elemental composition of CMB was analyzed using an organic elemental analyzer (FlashSmart, ThermoFisher Scientific, Republic of Korea).

  1. In conclusion, please avoid repeating the results again here. Just mention the most important ones and give your final suggestions based on the findings.

Response : Thanks for your valuable review and suggestions. We strongly believe that this study is valuable to get better understanding of how biochar impact on soil properties and crop production. This is particularly important for organic farming areas or carbon neutral agriculture. We completely agree to your suggestion. We modified conclusion section.

We really appreciate for the reviewer’s good comments to improve this manuscript.

I am looking forward to seeing your positive evaluation soon.

Best regard

Se-Won Kang

Round 2

Reviewer 2 Report

Comments and Suggestions for Authors

The Manuscript has been improved and potentially can be considered for publishing.

Author Response

We greatly appreciate the time and effort from the subject editor and two reviewers for reviewing our manuscript.